# Signal Quality Analysis of Single-Arm Electrocardiography

**DOI:** 10.3390/s23135818

**Published:** 2023-06-22

**Authors:** Jia-Jung Wang, Shing-Hong Liu, Cheng-Hsien Tsai, Ioannis Manousakas, Xin Zhu, Thung-Lip Lee

**Affiliations:** 1Department of Biomedical Engineering, I-Shou University, Kaohsiung 84001, Taiwan; wangjj@isu.edu.tw (J.-J.W.); charlieandfanny@yahoo.com.tw (C.-H.T.); manousakas@isu.edu.tw (I.M.); 2Department of Computer Science and Information Engineering, Chaoyang University of Technology, Taichung 413310, Taiwan; 3Division of Information Systems, School of Computer Science and Engineering, The University of Aizu, Aizu-Wakamatsu City 965-8580, Japan; zhuxin@u-aizu.ac.jp; 4Department of Cardiology, E-Da Hospital, Kaohsiung 84001, Taiwan; ed102601@edah.org.tw

**Keywords:** electrocardiogram (ECG), single-arm ECG, R-to-R wave interval (RRI), heart rate variability (HRV), signal-to-noise ratio (SNR)

## Abstract

The number of people experiencing mental stress or emotional dysfunction has increased since the onset of the COVID-19 pandemic, as many individuals have had to adapt their daily lives. Numerous studies have demonstrated that mental health disorders can pose a risk for certain diseases, and they are also closely associated with the problem of mental workload. Now, wearable devices and mobile health applications are being utilized to monitor and assess individuals’ mental health conditions on a daily basis using heart rate variability (HRV), typically measured by the R-to-R wave interval (RRI) of an electrocardiogram (ECG). However, portable or wearable ECG devices generally require two electrodes to perform bipolar limb leads, such as the Einthoven triangle. This study aims to develop a single-arm ECG measurement method, with lead I ECG serving as the gold standard. We conducted static and dynamic experiments to analyze the morphological performance and signal-to-noise ratio (SNR) of the single-arm ECG. Three morphological features were defined, RRI, the duration of the QRS complex wave, and the amplitude of the R wave. Thirty subjects participated in this study. The results indicated that RRI exhibited the highest cross-correlation (R = 0.9942) between the single-arm ECG and lead I ECG, while the duration of the QRS complex wave showed the weakest cross-correlation (R = 0.2201). The best SNR obtained was 26.1 ± 5.9 dB during the resting experiment, whereas the worst SNR was 12.5 ± 5.1 dB during the raising and lowering of the arm along the z-axis. This single-arm ECG measurement method offers easier operation compared to traditional ECG measurement techniques, making it applicable for HRV measurement and the detection of an irregular RRI.

## 1. Introduction

In contemporary society, job-related stress poses a significant issue due to its correlation with a substantial percentage (40–50%) of work absences [1]. Numerous previous studies have demonstrated that elevated levels of work-related stress can heighten the risk of various illnesses, including cardiovascular disease [2,3], neurodegenerative diseases [4], chronic aging-related ailments [5], and metabolic syndrome [6]. The concept of mental load acts as an intermediary between imposed demands and perceived demands. It also represents the disparity between the required capacities of the information processing system and the available capacities at any given time [1,7]. Evaluating mental workload has become essential to optimize work environments and enhance work performance [8]. Acker et al. have proposed a comprehensive framework that incorporates antecedent and moderating variables, as well as work-related consequences [9]. In this framework, the real-time measurement of individual physiological signals becomes crucial. Furthermore, by monitoring peoples’ mental conditions in real time, it is possible to identify and address unhealthy practices or poor working conditions to meet expectations [10,11].

While self-assessments such as the NASA-TLX questionnaire [12] can be used to evaluate mental workload, they are not without criticism [13]. In assessing mental workload through the physiological measurements, the methods such as pupillometry [14], electroencephalography [15], near-infrared spectroscopy [16], and heart rate monitoring [17] have gained popularity. With the universal use of wearable devices and mobile health applications (mHealth apps) in recent years, some tools have focused on objectively assessing mental health and neurological disorders in the home or work environment [11]. Additionally, the wearable devices have been developed to measure the various physiological signals, including electrocardiogram (ECG) patches [18,19], electromyogram (EMG) patches [20], pulse oxygen saturation (SpO2) [21], blood glucose [22], and light reflection rheogram (LRR) [23,24]. Among these, heart rate variability (HRV) has been found to have a strong correlation with mental activity, ease of acquisition [25], and sensitivity to changes in mental workload [26]. The R-to-R wave intervals (RRIs) of ECG can be applied to study the HRV. Liu et al. utilized a portable ECG device to measure real-time HRV for evaluating the mental workload of teachers in elementary schools [27,28]. However, current portable or wearable ECG devices require two electrodes to perform bipolar limb leads, such as the Einthoven triangle. This measurement setup can be challenging and uncomfortable for real-time HRV monitoring. The pulse rate variability (PRV) measured by the photoplethysmogram (PPG) has been studied in the last 10 years, and the PPG waveform is easily interfered with respiration, vascular characteristics, physical activity, and mental stress [29,30]. Thus, the PRV is an easy measurement indicator, but it cannot replace the HRV.

Traditionally, HRV has been viewed as a reflection of emotional response or stress. However, it is also an indicator of the body’s capacity to regulate internal and external demands. Various aspects of diet have been shown to have acute and long-term benefits for HRV. For instance, a Mediterranean diet, omega-3 fatty acids, B-vitamins, probiotics, polyphenols, and weight loss have all been associated with improved HRV. Conversely, dietary factors considered unfavorable, such as high intakes of saturated or trans fats and high glycemic carbohydrates, have been found to reduce HRV [31]. HRV has been found to be associated with levels of C-reactive protein (CRP) in the blood. CRP is produced by the liver in response to inflammation, with its levels increasing after the secretion of interleukin (IL)-6 from macrophages and T cells. Thus, Jarczok et al. have reported that the high-frequency (HF) component of HRV predicts the level of CRP [32]. Singh et al. have discovered a relationship between HRV and blood glucose levels [33]. The research literature has also explored the connection between HRV and micronutrients such as vitamins and minerals [34]. Thus, monitoring HRV in daily activities could be considered as the nutrient manager.

Therefore, daily HRV measurements are an important issue for self-care of mental health. To increase the measurement frequency, it is necessary to develop an ECG measurement method that is easily and comfortably used. The aim of this study was to develop a single-arm ECG measurement method, with the lead I ECG as the gold standard. We conducted an analysis of the morphological performance and signal-to-noise ratio (SNR) of the single-arm ECG under static and dynamic experiments. Three morphological features were defined: RRI, duration of QRS complex wave, and amplitude of the R wave. The static experiments included resting and Valsalva maneuver, while the dynamic experiments involved palm opening and clenching, forearm movement, horizontal arm movement along the *y*-axis, and raising and lowering of the arm along the *x*-axis and *z*-axis. A total of 30 subjects participated in the study. The results showed that RRI exhibited the highest cross-correlation (R = 0.9942) between the single-arm ECG and lead I ECG, whereas the duration of QRS complex wave showed the weakest cross-correlation (R = 0.2201). The best SNR was 26.1 ± 5.9 dB during the resting experiment, while the worst SNR was 12.5 ± 5.1 dB during the raising and lowering of the arm along the *z*-axis. Overall, the results indicated that RRI in the single-arm ECG had the strongest cross-correlation with the lead I ECG.

## 2. Materials and Methods

Figure 1 shows the framework of this study. A single-arm ECG measurement method was designed, and the obtained single-arm ECG morphology was compared with the lead I ECG being the golden standard under the static and dynamic experiments. The morphological parameters including the RRI, duration of QRS complex wave, and amplitude of the R wave were used to evaluate the performance of the QRS complex wave and SNR. The dynamic experiments consisted of five kinds of arm motions, while the static experiments contained the resting activity and Valsalva maneuver. In the study, the cross-correlation (R) and Bland–Altman plots were used to perform statistical analysis.

### 2.1. Single-Arm ECG

Figure 2 shows the block diagram of the single-arm ECG prototype. The averaged baseline of electrodes A and B were used to adjust the offset voltage of the instrument amplifier. The electrode C is the grounding with the power. The second-order inverting lowpass filter of the 3 Hz cutoff frequency extracted the baseline voltage to compensate the offset voltage of the instrument amplifier. The second-order high-pass filters of the 0.3 Hz cutoff frequency filtered the wandering baseline, the fourth-order low-pass filtered of the 40 Hz cutoff frequency filtered the high frequency noises, and the notch filter of the 60 Hz cutoff frequency filtered the noise of the power line. The total gain was 500 by the instrument amplifier and non-inverse amplifier. The instrumentation amplifier was AD620 (Analog Device, USA), and the operational amplifier was AD082 (Analog Device, Norwood city, OH, USA). All filters were implemented with the Butterworth structure. The noise of the single-arm ECG prototype was the output signal when the terminals of electrodes A and B were connected to the ground of this device. Then, the amplitudes of the R wave and noise were used to evaluate the SNR. The SNR of this prototype was 26.1 ± 5.1 dB. The data acquisition system was the Biopac MP150 (BioPack Systems Inc. MP150). The lead I ECG was measured using the ECG100C ECG amplifier module (BioPack Systems Inc., Goleta, CA, USA). The sampling frequency was 1 kHz.

### 2.2. Protocol of Experiment

This study recruited 30 young subjects (30 males) without cardiovascular disease or injured limbs. Their age was between 20 and 23 years (22.1 ± 0.9 years, mean ± standard deviation, SD), their weight was between 51 and 78 Kg (61.8 ± 9.2 Kg), and their height was between 161 and 182 cm (172.1 ± 8.9 cm). This experiment was approved by the Institutional Review Board of the E-DA Hospital, Kaohsiung, Taiwan (No. EMRP-111-013), and informed consent was acquired from each participant prior to initiation of the study.

Three electrodes of the single-arm ECG are put at the left upper arm near the shoulder, as shown in Figure 3. However, three electrodes of the lead I ECG were placed at the left and right hands and right leg, respectively. When the electrodes of the single-arm ECG were placed on the arm, EMG was the noise that interfered the most with such ECG. Thus, the experiments were designed to perform the arm motions in the *x*-axis, *y*-axis, *z*-axis, and forearm motion. Moreover, since the EMG noise did affect both genders, only the male subjects were recruited in this study. Then, the morphological performance and SNRs of the single-arm ECG were evaluated under the five dynamic and two static experiments. The static experiments included the resting activity and Valsalva maneuver. The dynamic experiments were the palm opening and clenching, forearm moving, arm horizontally moving along the *y*-axis, arm raising and lowering along the *x*-axis, and arm raising and lowering along the *z*-axis. These arm motions were related to the activities of many muscles. The major muscles comprise the trapezius, deltoid, pectoralis major, biceps brachii, and triceps brachii. Table 1 shows the seven experimental protocols. Figure 4 shows the schematic diagrams of the arm motions along the *x*-, *y*-, and *z*-axes. Figure 4a is the arm horizontally moving along the *y*-axis, Figure 4b the arm raising and lowering along the *x*-axis, and Figure 4c the arm raising and lowering along the *z*-axis.

### 2.3. Extraction of ECG Parameters

The single-arm ECG and lead I ECG signals were filtered to remove the wandering noise in baseline and the high frequency noise using a fourth-order Butterworth bandpass filter in which the lower and upper cutoff frequencies were 0.5 Hz and 40 Hz, respectively. To reduce the differences of phase lag among different signals, an eight-order all-pass filter was designed to equalize the group delay within the passband. The R waves of the single-arm ECG and lead I ECG were detected using the Pan–Tompkins method [35]. Figure 5 shows the single-arm ECG and Lead I ECG in the resting activity. The QRS complex waves have the larger distortion for the single-arm ECG than the lead I ECG. However, the R waves are clear and sharp for the two ECG signals. Figure 6 shows a one-beat waveform of the single-arm ECG with the lowest distortion. The P, Q, R, S and T waves are roughly marked. In this study, the interval of R-to-R waves (RRI), duration of QRS complex wave, and amplitude of R wave were used to evaluate the morphological performance of the single-arm ECG. The Q and S waves are the left- and right-first valleys of R wave, and the amplitude of R wave is the amplitude of the R to Q waves.

### 2.4. Statistical Analysis

The quantitative data are expressed as the mean ± standard deviation (SD). A two-tailed paired t-test is used to show the difference of two variables. A *p*-value of 0.05 or less is considered statistically significant. The cross-correlation coefficient, *R*, is the quantity that gives the quality of the least squares fitting to the original data:(1)$$R=\sqrt{\frac{{\left[n(\sum xy)-(\sum x)(\sum y)\right]}^{2}}{[n\sum x^{2}-{\left(\sum x\right)}^{2}][n\sum y^{2}-{(\sum y)}^{2}]}}$$

Furthermore, the precision and agreement between the single-arm ECG and lead I ECG are compared using a Bland–Altman plot.

## 3. Results

According to the static and dynamic experiments, we evaluated the SNRs of these single-arm ECG signals and compared them with the SNR under the resting experiment. The ECG noise was extracted from the T wave to the next *p* wave. The maximum amplitude in these intervals was defined as the noisy amplitude. Table 2 shows the SNRs under the seven experiments. In the dynamic experiment, the palm opening and clenching experiment has the maximum SNR, 23.5 ± 7.4 dB, and the arm raising and lowering along the *z*-axis experiment has the minimum SNR, 12.5 ± 5.1 dB. In the static experiments, these SNRs do not have significant a difference, 26.1 ± 5.9 dB (resting activity) vs. 23.0 ± 7.4 dB (Valsalva maneuver).

The RRIs of the single-arm ECG and lead I ECG were compared with cross-correlation under the seven experiments, as shown in Table 3. The numbers of RRI in the resting activity (30 s), palm opening and clenching (30 s), forearm moving (30 s), arm horizontally moving along the *y*-axis (30 s), arm raising and lowering along the *x*-axis (30 s), arm raising and lowering along the *z*-axis (30 s), and Valsalva maneuver (70 s) were 1174, 1138, 1147, 1150, 1149, 1129, and 2681, respectively. Therefore, the number of total RRIs was 9568. Figure 7 shows the cross-correlation of all RRIs between the two kinds of ECG signals, with a coefficient of 0.9942. In addition, the cross-correlation coefficients for all subjects under the seven experiments are almost greater than 0.970. Thus, a good linear relationship existed between RRIs extracted from the single-arm ECG and the lead I ECG. Figure 8 shows the Bland–Altman plot of all RRIs (9568 pairs). In the figure, the mean of difference between the two RRIs, respectively, extracted from the single-arm and lead I ECGs is found to be close to 0.0 ms, and the upper and lower limitations of agreement are 11.8 ms and −11.8 ms, respectively. Since 95.3% of the RRI pairs are within the agreement interval, it implies that a good agreement exists between the RRI pairs from these two ECG signals.

Moreover, the durations of the QRS complex waves from these two ECG signals were compared. Table 4 shows the cross-correlation coefficients under the seven different experiments. It was found that the best cross-correlation coefficient (0.239 ± 0.117) for the QRS durations was obtained in the resting activity, and the worst coefficient (0.172 ± 0.132) in the Valsalva maneuver activity. Figure 9 shows a cross-correlation coefficient of 0.2201 for all durations of the QRS complex waves in the seven experiments. Consequently, there was only a low relation between the duration of the QRS complex waves extracted from these two ECGs.

Finally, the amplitude of R wave for these two ECG signals were compared. Table 5 shows the cross-correlation coefficients under the seven experiments. The resting activity and the arm raising and lowering in the *x*-axis experiment results in the best and worst cross-correlation coefficients (0.5096 ± 0.2368, 0.2182 ± 0.1777). Figure 10 shows a cross-correlation coefficient of 0.4258 for all amplitudes of R waves extracted under the seven experiments. Accordingly, the amplitude of R wave from the single-arm ECG moderately correlates with that from the lead I ECG.

## 4. Discussion

In the study, the main needs for the development of a single-arm ECG measurement device include its ease of use, long-term monitoring, and wearability. In the lead I ECG configuration, two electrodes placed at two arms are used to measure a body surface’s half-cell potentials, and a third electrode placed at the right leg provides a low-impedance return path to the ground for noise reduction. A commercial ECG apparatus (ECG100C, BioPack Systems Inc., Goleta, CA, USA) was used to measure the single-arm ECG. Unfortunately, its signal owned very small amplitude and was interfered with by power-line noise and the motion artifacts. However, the proposed single-arm ECG device, as shown in Figure 2, amplifies the minute half-cell potentials, usually less than 0.1 mV, between the two electrodes on the upper arm. The average potential of two electrodes with high-resistance resistors (R1) was used as the neutral potential to shield the lead wires of two electrodes and the reference of instrument amplifier by the resistors (R_G_/2). This design could effectively reduce the difference between the two input commons. In addition, high-order low-pass filters were incorporated to filter the frequency components above the 40 Hz and then to increase the signal-to-noise ratio.

The quality of the single-arm ECG signal is lower compared to the lead I ECG, and this is influenced by the positioning of the electrodes. Figure 11a illustrates a single-arm ECG during the resting activity, where the single-arm ECG quality is relatively high. However, when the subject performs the horizontal arm movement along the *y*-axis, continuous contraction of the deltoid muscle results in a larger amount of EMG coupling with the ECG signal, as shown in Figure 11b. Consequently, the ECG quality becomes deteriorated. To address this issue, the electrode positions are adjusted based on the subject’s arm structure, such as the arm length, muscle size, and arm circumference. Figure 11c shows the single-arm ECG after electrode adjustment, where the R wave is clearly displayed. Table 2 shows that when subjects were practicing the arm moving horizontally along the *y*-axis, arm raising and lowering along the *x*-axis, and arm raising and lowering along the *z*-axis the SNRs are the lowest, measuring at 13.1 ± 3.8 dB, 12.6 ± 5.5 dB, and 12.5 ± 5.1 dB, respectively. In most subjects, their SNRs in the resting activity are significantly higher than those in the dynamic activities, while some subjects have lower SNRs in the resting activity than those in the dynamic activities, such as subjects 1 and 25. The major reason is that the half-cell potentials of the two electrodes are not close to drop the common-mode rejection ratio (CMRR) of the instrument amplifier. This problem will be overcome by the circuit design in the future when the single-arm ECG is developed as a fitness band.

In this study, the signal quality of the single-arm ECG was examined by evaluating three morphological parameters, including the RRI, duration of the QRS complex wave, and amplitude of the R wave. These parameters between the single-arm and lead I ECGs were compared, and their cross-correlation coefficients were found to be 0.9942, 0.2201, and 0.4258, respectively, as shown in Figure 7, Figure 9 and Figure 10. Figure 5 illustrates the waveform of the single-arm ECG, which exhibits a distinct R wave while lacking P and T waves. The Q and S waves are not clearly discernible. Consequently, the RRI demonstrates the highest cross-correlation coefficient of 0.9942, while the duration of the QRS complex wave displays the lowest cross-correlation coefficient of 0.2201. The error in RRI can be attributed to two reasons. Firstly, the single-arm ECG experiences greater baseline drift compared to the lead I ECG, as evident in Figure 12, when the subjects perform arm motions. Secondly, the nonlinear phase characteristics of Butterworth filters, used as high-pass (cutoff frequency: 0.5 Hz) and low-pass (cutoff frequency: 40 Hz) filters, and the infinite impulse response (IIR) filters contribute to this error. Moreover, the errors observed in the duration of the QRS complex wave and amplitude of the R wave are caused by the presence of coupled EMG, as shown in Figure 13. The Q and S waves, coupled with the EMG, lack a clear and well-defined waveform. Table 2 presents the SNRs of the single-arm ECG under different experiments. The greater the arm motion intensity, the lower the SNR of the single-arm ECG.

According to results of this study, the single-arm ECG shows promising potential for development as a wearable HRV device, primarily due to its high accuracy in measuring RRI. Furthermore, major technology companies like Apple and Google have recently introduced health applications such as Apple Health and Google Fit, which incorporate ECG measurement and arrhythmia detection based on RRI [36]. These applications are compatible with their respective smartwatches, where users are required to place their finger on the crown of the watch during ECG measurements [37]. However, their measurement method may not be comfortable or user-friendly for long-term monitoring. The traditional Holter ECG and modern ECG patch [19] are worn on the chest to continuously measure the ECG for the diagnosis of heart disease. When users are wearing or taking off it, their clothes must be removed. Moreover, the electrodes of the Holter monitors are arranged on the chest, in which motion artifacts are induced by respiration and cardiac beating. Obviously, both low frequency noise (<0.2 Hz) caused by the respiration activity and that (about 1.2 Hz) caused by the cardiac movement is superimposed on the Holter ECG waveform [38]. Now, the single-arm ECG device is a prototype. Thus, its wearing method could not easier than the Holter ECG and ECG patch. In the future, it has great potential to be developed as a fitness band worn on the arm to monitor regular and irregular heartbeats. Thus, its advantages will be that its wearing method could be more convenient than the Holter ECG and ECG patch, and its measuring time could be longer than the Apple watch.

The primary limitation of the single-arm ECG is the placement of electrodes only on the left arm. We have not investigated whether other electrode positions, such as the right hand or left leg, are suitable or not. Additionally, due to the presence of significant noise, such as EMG interference, the Q and S waves of the single-arm ECG do not exhibit clear and well-defined waveforms. It is only the RRI parameter that demonstrates high accuracy, thus restricting its broader application. To address this limitation, various digital signal-processing techniques can be employed to mitigate artifact noise in the ECG. Methods such as adaptive filtering [39], empirical mode decomposition [40], or principal component analysis [41] could be utilized to effectively remove unwanted noise.

## 5. Conclusions

In this study, we extracted RRI, duration of QRS complex wave, and amplitude of R wave from the single-arm ECG and compared them with the corresponding parameters obtained from the lead I ECG. The resulting cross-correlation coefficients were 0.9942, 0.2201, and 0.4258, respectively. The measurement method employed in this study offers a more straightforward operation compared to traditional ECG measurement methods, making it suitable for HRV measurement and for monitoring regular and irregular heartbeats. Now, the single-arm ECG in this study is only a prototype that needs lead cables. In the future, it could be designed as a fitness band, and several experiments related to daily activities will be performed to verify its performance.

## Figures and Tables

**Figure 1 sensors-23-05818-f001:**
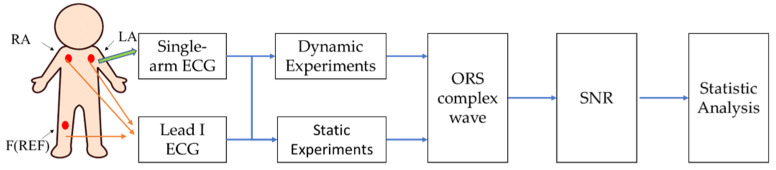
The framework of this study. The lead I ECG as the golden standard is used to evaluate performance of the single-arm ECG under dynamic and static experiments.

**Figure 2 sensors-23-05818-f002:**
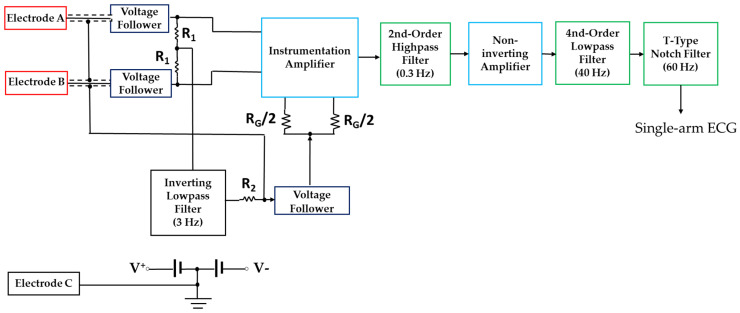
Block diagrams of circuits for the single-arm ECG measurement.

**Figure 3 sensors-23-05818-f003:**
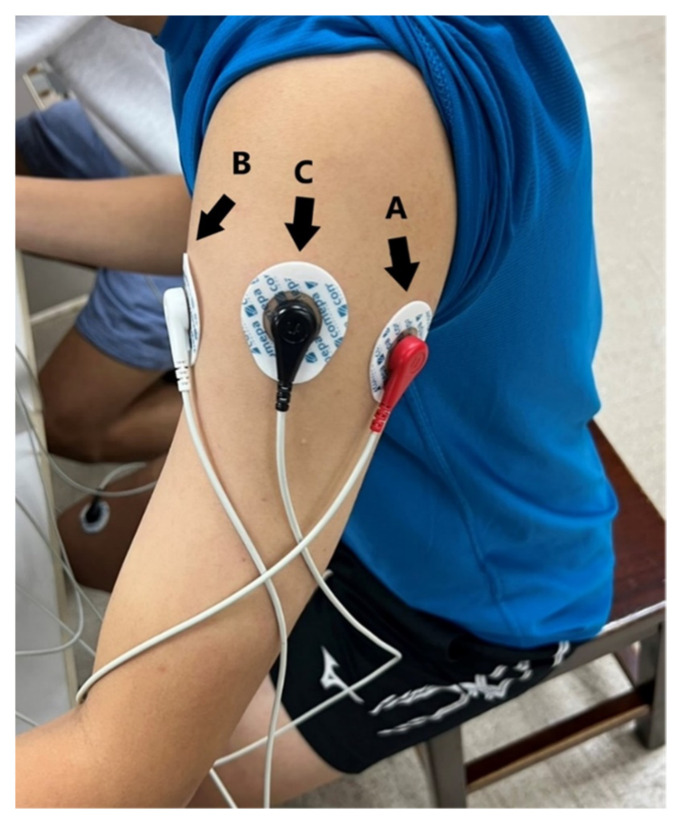
Position of the three electrodes in the single-arm ECG prototype. The electrodes in Figure 2 are placed at the shoulder, which are sorted by A, C, and B from right to left.

**Figure 4 sensors-23-05818-f004:**
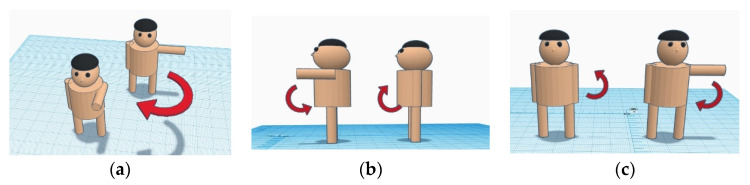
The schematic diagrams of arm motions along the *x*-, *y*-, and *z*-axes, (**a**) arm horizon-tally moving along the *y*-axis, (**b**) arm raising and lowering along the *x*-axis, (**c**) arm raising and lowering along the *z*-axis.

**Figure 5 sensors-23-05818-f005:**
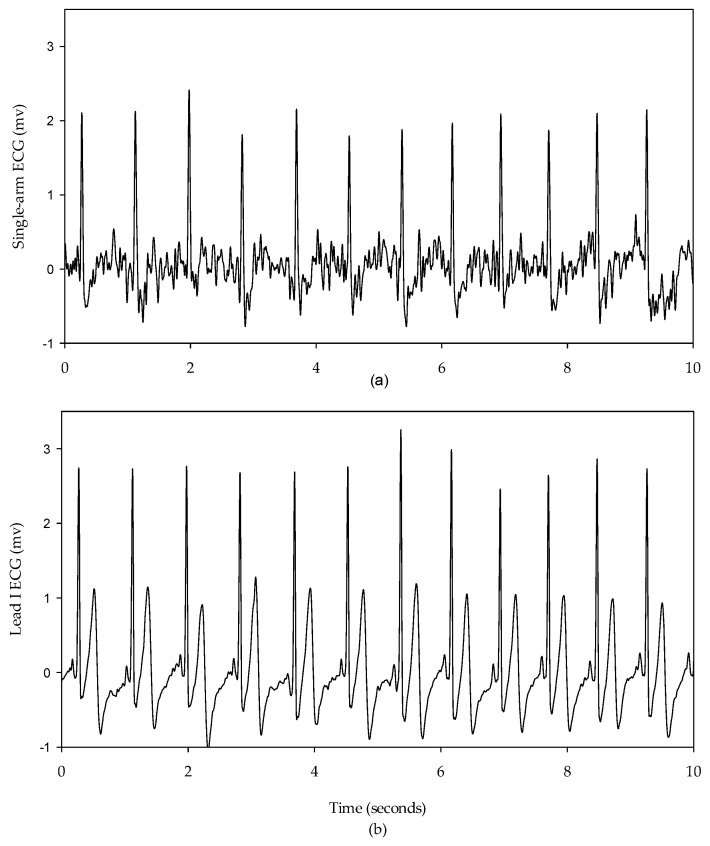
Subject is at resting activity, (**a**) single-arm ECG, (**b**) lead I ECG.

**Figure 6 sensors-23-05818-f006:**
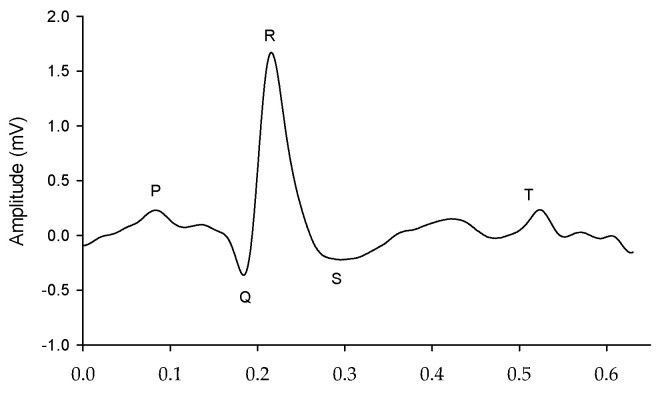
One-beat waveform of the single-arm ECG.

**Figure 7 sensors-23-05818-f007:**
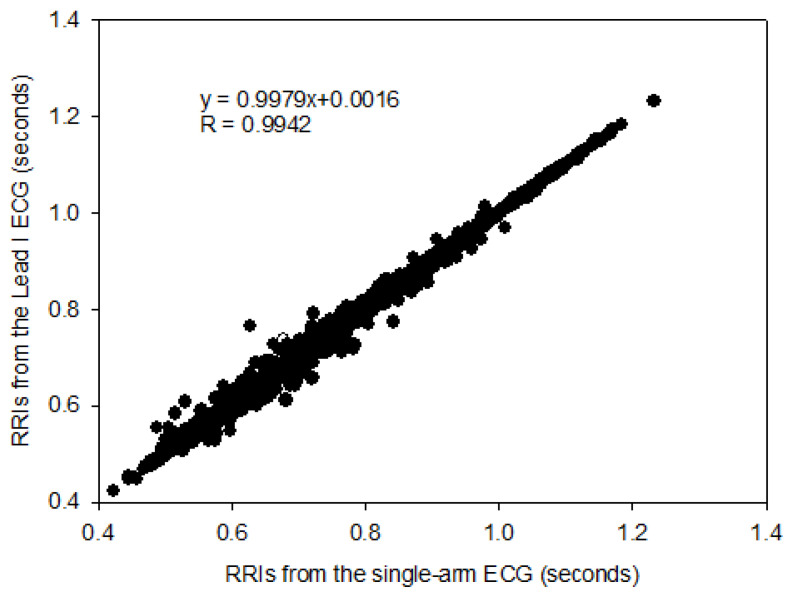
The cross-correlation of all RR intervals, respectively, extracted from the single-arm ECG and lead I ECG.

**Figure 8 sensors-23-05818-f008:**
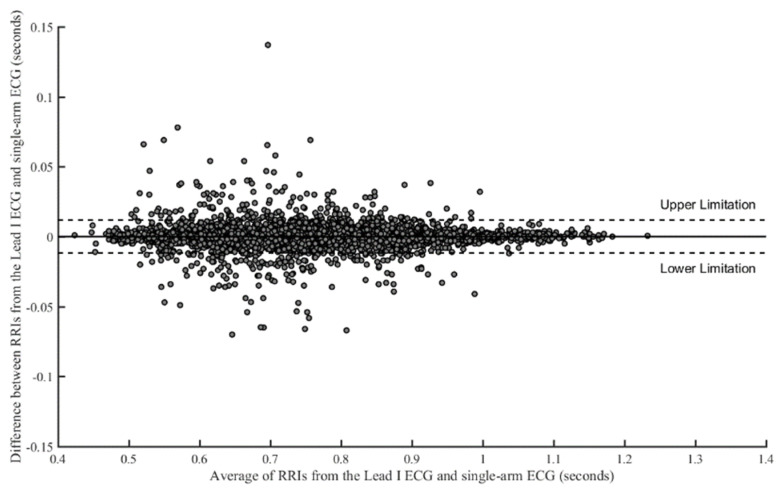
Bland–Altman plot of all RRIs. The mean is close to 0.0 ms, and upper and lower limitations of agreement are 11.8 ms and −11.8 ms, respectively. There are 95.3% of RRIs within the agreement interval.

**Figure 9 sensors-23-05818-f009:**
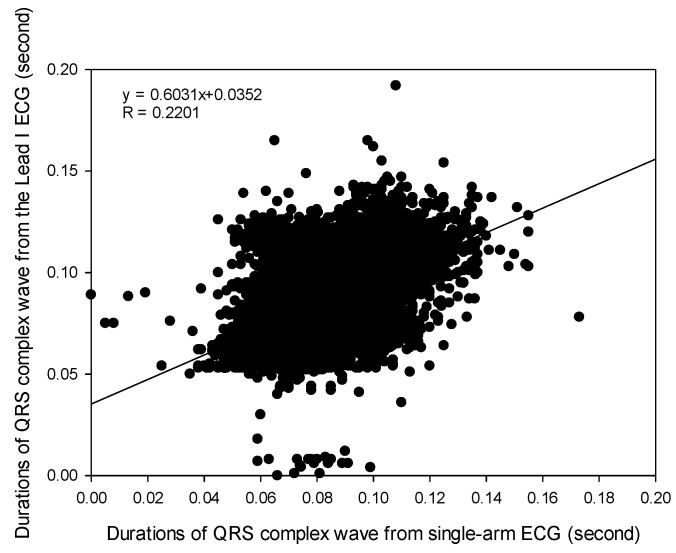
The cross-correlation of all durations of QRS complex waves, respectively, extracted from the single-arm ECG and lead I ECG.

**Figure 10 sensors-23-05818-f010:**
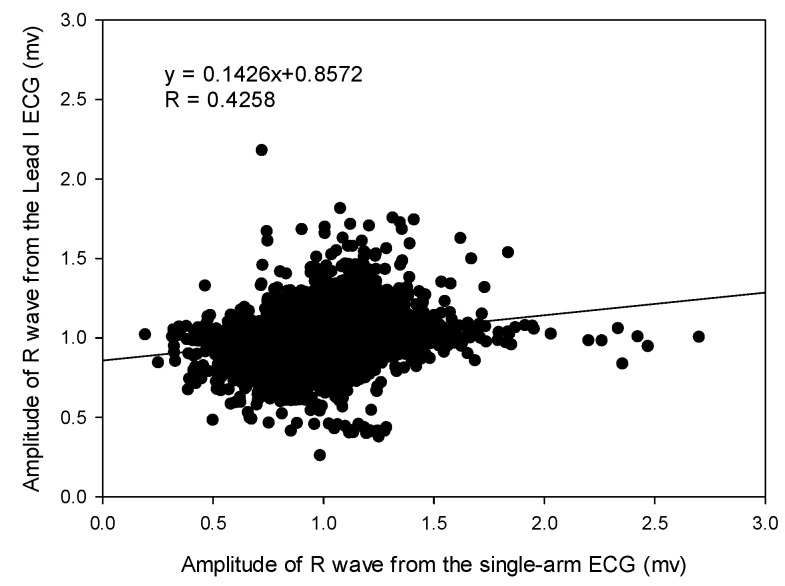
The cross-correlation of all amplitudes of R waves, respectively, extracted from single-arm ECG and lead I ECG.

**Figure 11 sensors-23-05818-f011:**
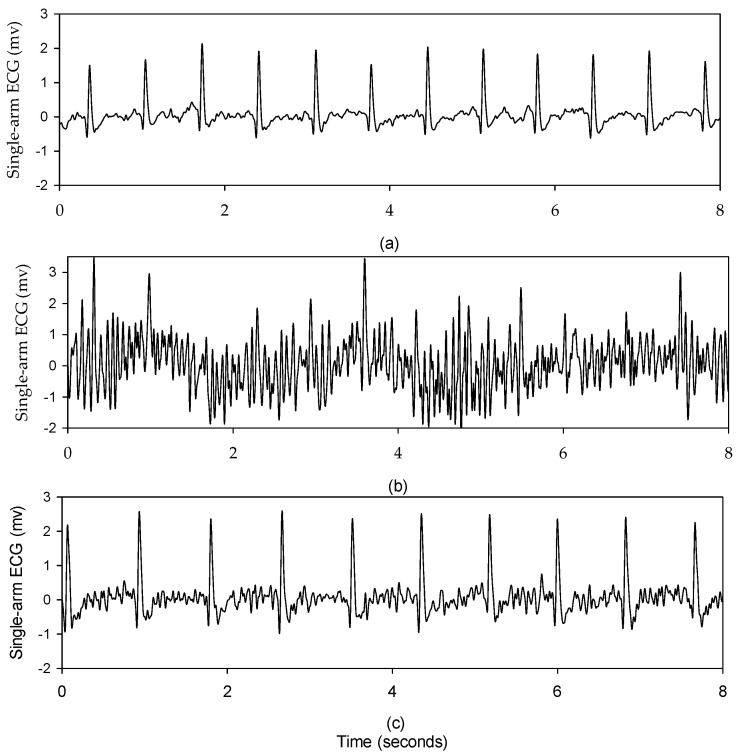
The single-arm ECG of one subject under the different experiments, (**a**) the resting activity, (**b**) the experiment conducted with the arm horizontally moving along the *y*-axis, (**c**) after electrodes adjustment, and the experiment conducted with the arm horizontally moving along the *y*-axis experiment.

**Figure 12 sensors-23-05818-f012:**
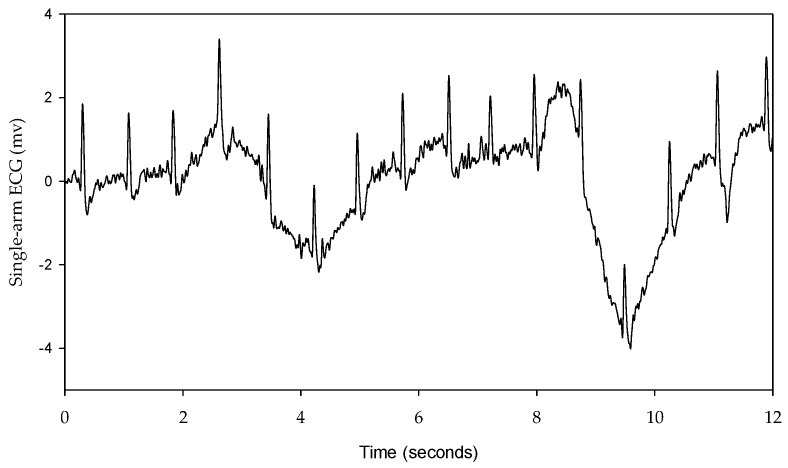
The single-arm ECG exhibits baseline drift when the subject performs the arm movements.

**Figure 13 sensors-23-05818-f013:**
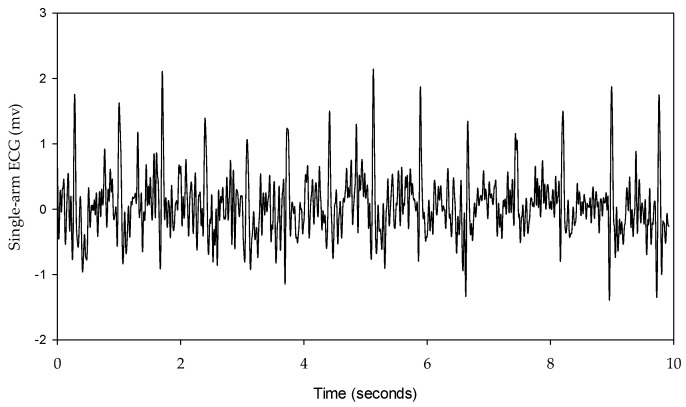
The single-arm ECG signal is affected by the EMG activity during arm movements.

**Table 1 sensors-23-05818-t001:** Seven experimental protocols.

No.	Experiment	Protocol
1	Resting	Subjects in a sitting position were asked to take a rest at least three minutes to allow their hemodynamic status to become stable before the ECG measurement began. Subjects put their two arms on a table relaxedly. In the resting activity, 30 s single-arm and lead I ECGs were simultaneously registered.
2	Palm opening and clenching	Subject were asked to put their arms on the table, stretch their arms, and continuously open and clench their left palm 10 times. Each opening and clenching of the palm lasted 3 s. Then, subjects were asked to take a rest for three minutes.
3	Forearm moving	Subject were asked to put their arms on the table, stretch their arms, and raise their left forearm from 0° to 90° and 90° to 0° 10 times. Each movement lasted 3 s. Then, subjects were asked to take a rest for three minutes.
4	Arm horizontally moving along *y*-axis	Subjects in a standing position were asked to raise their left arm to 90° and horizontally move their arm 10 times. The moving angle was 90°. Each movement lasted 3 s. Then, subjects were asked to take a rest for three minutes.
5	Arm raising and lowering along *x*-axis	Subjects in a standing position were asked to raise their left arm and lower their arm in the horizontal direction 10 times. The moving angle was 90°. Each movement lasted 3 s. Then, subjects were asked to take a rest for three minutes.
6	Arm raising and lowering along *z*-axis	Subjects in a standing position were asked to raise their left arm and lower their arm in the vertical direction 10 times. The moving angle was 90°. Each movement lasted 3 s. Then, subjects were asked to take a rest for three minutes.
7	Valsalva maneuver	Subjects performed the Valsalva maneuver by blowing into a rubber tube connected to a mercury column sphygmomanometer and maintaining a pressure of 50 mm Hg for 10 s. Then, subjects were asked to take a rest for three minutes.

**Table 2 sensors-23-05818-t002:** SNRs of the single-arm ECG recorded for 30 subjects under seven experiments.

Sub.	Resting	Palm Opening and Clenching	Forearm Moving	Arm Horizontally Moving in *y*-Axis	Arm Raising and Lowering in *x*-Axis	Arm Raising and Lowering in *z*-Axis	Valsalva Maneuver
1	5.6	4.1	5.8	4.053	4.053	4.023	6.120
2	11.0	14.4	8.8	3.783	2.918	2.780	31.740
3	45.2	35.0	15.5	3.805	4.282	3.976	24.733
4	23.6	38.6	7.8	3.633	2.416	2.198	34.865
5	25.7	17.7	12.0	5.812	5.338	5.726	8.203
6	23.8	19.5	6.4	2.499	3.466	2.677	26.281
7	79.3	26.8	8.7	10.351	2.014	7.185	12.603
8	17.3	28.6	16.5	5.063	4.892	5.602	12.390
9	46.8	4.4	5.9	4.358	3.984	4.520	19.101
10	21.4	17.9	6.4	6.647	7.198	4.574	28.793
11	20.7	166.1	21.2	3.875	8.343	4.303	15.002
12	24.5	8.3	10.622	2.591	2.747	2.022	8.545
13	59.0	39.5	36.921	8.536	11.209	6.967	34.430
14	34.2	7.4	6.976	2.049	2.256	2.162	17.363
15	17.6	10.0	3.893	5.497	6.829	4.822	6.820
16	20.7	15.8	5.778	5.754	9.902	7.586	13.984
17	19.9	27.6	7.913	7.454	19.066	52.994	16.338
18	16.1	6.3	10.957	4.433	1.869	2.095	6.386
19	58.9	23.5	30.860	13.050	6.246	4.917	46.567
20	10.1	42.8	11.698	5.628	5.058	5.960	12.646
21	16.0	20.9	13.408	6.294	6.306	6.426	20.706
22	13.3	9.8	8.118	7.815	4.186	6.109	7.315
23	12.8	30.2	6.869	7.852	8.319	7.893	16.724
24	10.9	12.1	3.178	2.521	2.761	3.113	9.533
25	9.5	25.7	4.705	7.336	6.933	5.008	20.319
26	49.6	17.0	15.197	3.253	2.172	2.622	48.128
27	24.3	12.9	7.831	9.014	6.238	7.089	18.459
28	14.7	7.7	14.124	5.397	3.550	4.790	18.746
29	56.0	67.0	31.940	6.612	4.984	5.513	34.386
30	20.0	18.3	5.285	4.498	3.312	2.409	13.245
Mean ± SD (dB)	26.1 ± 5.9	23.5 ± 7.4	18.3 ± 5.5 **	13.1 ± 3.8 **	12.6 ± 5.5 **	12.5 ± 5.1 **	23.0 ± 7.4

**: *p* < 0.001.

**Table 3 sensors-23-05818-t003:** Cross-correlation of RRI, respectively, extracted from the single-arm ECG and lead I ECG in 30 subjects under seven experiments.

Sub.	Resting	Palm Opening and Clenching	Forearm Moving	Arm Horizontally Moving in *y*-Axis	Arm Raising and Lowering in *x*-Axis	Arm Raising and Lowering in *z*-Axis	Valsalva Maneuver
1	0.9980	0.9881	0.9961	0.9919	0.9986	0.9971	0.9994
2	0.9992	0.9982	0.9983	0.9909	0.9944	0.9979	0.9996
3	0.9991	0.9964	0.9975	0.9699	0.9655	0.9222	0.9998
4	0.9983	0.9997	0.9977	0.9812	0.9760	0.8991	0.9599
5	0.9477	0.9920	0.9997	0.9863	0.9908	0.9992	0.9992
6	0.9994	0.9998	0.9986	0.8852	0.9954	0.9918	0.9199
7	0.9671	0.9997	0.9956	0.9983	0.9505	0.9991	0.9998
8	0.9964	0.9990	0.9987	0.9878	0.9977	0.9887	0.9997
9	0.9997	0.9980	0.9968	0.9970	0.9985	0.9974	0.9998
10	0.9989	0.9992	0.9977	0.9973	0.9987	0.9975	0.9995
11	0.9989	0.9985	0.9958	0.9865	0.9758	0.9712	0.9996
12	0.9998	0.9953	0.9974	0.9521	0.9544	0.9550	0.9599
13	0.9996	0.9978	0.9770	0.9934	0.9812	0.9989	0.9998
14	0.9994	0.9968	0.9967	0.8561	0.8207	0.8054	0.9957
15	0.9969	0.8431	0.8867	0.9764	0.9869	0.9829	0.9991
16	0.9998	0.9993	0.9966	0.9894	0.9987	0.9990	0.9997
17	0.9975	0.9997	0.9952	0.9796	0.9554	0.9438	0.9999
18	0.9994	0.9837	0.9981	0.9953	0.9413	0.9137	0.9918
19	0.9991	0.9948	0.9945	0.9981	0.9759	0.9527	0.9998
20	0.9970	0.9972	0.9978	0.9713	0.9633	0.9913	0.9948
21	0.9993	0.9962	0.9975	0.9978	0.9926	0.9945	0.9997
22	0.9998	0.9999	0.9991	0.9996	0.9985	0.9992	0.9994
23	0.9984	0.9986	0.9966	0.9997	0.9996	0.9987	0.9997
24	0.9995	0.9989	0.9753	0.9833	0.9836	0.9968	0.9799
25	0.9988	0.9991	0.9962	0.9973	0.9975	0.9960	0.9996
26	0.9996	0.9992	0.9994	0.9953	0.9198	0.9687	0.9995
27	0.9990	0.9987	0.9980	0.9982	0.9494	0.9917	0.9998
28	0.9996	0.9988	0.9994	0.9881	0.9941	0.9368	0.9997
29	0.9995	0.9977	0.9991	0.9898	0.9823	0.9997	0.9999
30	0.9997	0.9998	0.9997	0.9980	0.9984	0.9963	0.9996
Mean ± SD	0.996 ± 0.012	0.992 ± 0.028	0.992 ± 0.0217	0.981 ± 0.032	0.974 ± 0.036	0.973 ± 0.043	0.993 ± 0.017

**Table 4 sensors-23-05818-t004:** Cross-correlation of durations of QRS complex waves, respectively, extracted from the single-arm and lead I ECGs in 30 subjects under seven experiments.

Sub.	Resting	Palm Opening and Clenching	Forearm Moving	Arm Horizontally Moving in *y*-Axis	Arm Raising and Lowering in *x*-Axis	Arm Raising and Lowering in *z*-Axis	Valsalva Maneuver
1	0.1375	0.3293	0.0400	0.2205	0.2280	0.1353	0.0283
2	0.2159	0.1729	0.0600	0.3059	0.1490	0.2307	0.0202
3	0.1732	0.0100	0.3869	0.2665	0.3437	0.1140	0.1389
4	0.2114	0.2490	0.0100	0.0663	0.0714	0.0469	0.1411
5	0.5119	0.4051	0.3200	0.4848	0.0794	0.0300	0.3074
6	0.1253	0.3176	0.1237	0.4601	0.0781	0.1257	0.1233
7	0.2791	0.0943	0.5001	0.0003	0.3490	0.2528	0.1371
8	0.3162	0.2149	0.1761	0.1552	0.0843	0.1049	0.3082
9	0.3030	0.3777	0.2258	0.2848	0.0557	0.1490	0.0975
10	0.1539	0.1676	0.1296	0.3709	0.1720	0.2012	0.0283
11	0.1068	0.2784	0.4156	0.4413	0.3338	0.3719	0.4030
12	0.3361	0.0374	0.0361	0.2390	0.0922	0.3053	0.1356
13	0.3897	0.2352	0.1459	0.1020	0.5039	0.1265	0.1552
14	0.2114	0.0361	0.0678	0.2798	0.1847	0.0469	0.3592
15	0.1872	0.2274	0.0447	0.1237	0.1817	0.2324	0.0959
16	0.3856	0.0608	0.1407	0.3158	0.0775	0.2828	0.0080
17	0.1707	0.3098	0.1634	0.0020	0.1884	0.0985	0.2583
18	0.1224	0.0141	0.1414	0.0917	0.1520	0.1700	0.0843
19	0.1712	0.3966	0.0374	0.0245	0.2655	0.0800	0.1153
20	0.2015	0.2315	0.3214	0.2140	0.2377	0.0608	0.0883
21	0.5217	0.0520	0.3581	0.0245	0.1910	0.5675	0.2561
22	0.1794	0.2615	0.4159	0.4287	0.0911	0.0894	0.1221
23	0.4643	0.1245	0.3941	0.2128	0.0283	0.4315	0.4845
24	0.2914	0.2746	0.3908	0.1058	0.0173	0.4887	0.0265
25	0.1281	0.0877	0.0224	0.0265	0.2782	0.0566	0.2364
26	0.1342	0.2458	0.3373	0.2285	0.4591	0.5993	0.1876
27	0.1332	0.5253	0.4438	0.6226	0.5657	0.4860	0.4874
28	0.2374	0.6227	0.2651	0.5542	0.3685	0.5631	0.1253
29	0.1456	0.0566	0.2184	0.4426	0.5031	0.3350	0.0412
30	0.2236	0.2742	0.0592	0.0300	0.0436	0.1396	0.1825
Mean ± SD	0.240 ± 0.118	0.223 ± 0.152	0.213 ± 0.152	0.237 ± 0.177	0.212 ± 0.154	0.230 ± 0.174	0.172 ± 0.132

**Table 5 sensors-23-05818-t005:** Cross-correlation of R-wave amplitudes, respectively, extracted from the single-arm ECG and lead I ECG in 30 subjects under seven experiments.

Sub.	Resting	Palm Opening and Clenching	Forearm Moving	Arm Horizontally Moving in *y*-Axis	Arm Raising and Lowering in *x*-Axis	Arm RAISING and lowering in *z*-Axis	Valsalva Maneuver
1	0.4488	0.1954	0.3268	0.3278	0.1841	0.2020	0.2666
2	0.1208	0.2309	0.1005	0.1304	0.1217	0.3913	0.2951
3	0.2818	0.2874	0.0424	0.0010	0.2508	0.2581	0.4268
4	0.6623	0.3713	0.0721	0.0700	0.0346	0.0781	0.7014
5	0.1179	0.6220	0.2683	0.1241	0.1367	0.0500	0.5119
6	0.6719	0.2888	0.2888	0.1822	0.0800	0.2860	0.7794
7	0.1082	0.7096	0.3747	0.3803	0.0100	0.3363	0.2293
8	0.3030	0.0021	0.2846	0.1549	0.0361	0.0374	0.5247
9	0.7752	0.5646	0.3429	0.0469	0.4204	0.6111	0.2121
10	0.3000	0.3822	0.3191	0.0245	0.3151	0.3146	0.3291
11	0.3288	0.1360	0.2919	0.0300	0.0566	0.1775	0.0938
12	0.5625	0.1196	0.1058	0.0224	0.0011	0.0412	0.7246
13	0.6738	0.8506	0.8476	0.1572	0.3685	0.3211	0.6346
14	0.1812	0.5358	0.4584	0.0825	0.0548	0.1552	0.3041
15	0.4252	0.4431	0.0100	0.2843	0.3453	0.3651	0.4535
16	0.1954	0.5815	0.3370	0.0012	0.4747	0.2177	0.6416
17	0.4004	0.3872	0.0721	0.1616	0.0529	0.2980	0.5500
18	0.7578	0.6494	0.7075	0.6567	0.3186	0.2522	0.6001
19	0.5525	0.8440	0.4709	0.1885	0.2555	0.3308	0.0141
20	0.7233	0.7394	0.8180	0.3821	0.2443	0.2289	0.4768
21	0.6351	0.7394	0.5617	0.5246	0.2707	0.0837	0.5607
22	0.6764	0.6393	0.2828	0.4498	0.1349	0.6754	0.0648
23	0.7928	0.1640	0.5504	0.4491	0.4792	0.5006	0.1241
24	0.8514	0.3966	0.2339	0.2121	0.2076	0.2644	0.1249
25	0.6657	0.7745	0.5712	0.6358	0.7125	0.5086	0.6938
26	0.4675	0.3367	0.2298	0.1367	0.1149	0.4300	0.4473
27	0.3784	0.1304	0.2835	0.1606	0.0012	0.0592	0.2400
28	0.8011	0.5960	0.7459	0.2610	0.1543	0.0436	0.9011
29	0.5602	0.5819	0.5059	0.0917	0.1931	0.3085	0.8012
30	0.8691	0.9487	0.6322	0.5982	0.5168	0.4812	0.8725
Mean ± SD	0.509 ± 0.236	0.475 ± 0.253	0.371 ± 0.230	0.230 ± 0.195	0.218 ± 0.177	0.276 ± 0.171	0.453 ± 0.253

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
