# Peer review of "Signal Quality Analysis of Single-Arm Electrocardiography"

_sensors, 2023, doi:10.3390/s23135818_

Round 1

Reviewer 1 Report

This paper presents the development of a single-arm ECG device for the continuous monitoring of heart for mental health assessment purposes. The motivation for the work is borne out of a need for an ECG measurement setup that is comfortable and easy to use, and thus be suitable for the continuous monitoring of heart-related conditions. The performance of the device and measurement setup is compared against Lead I ECG recording using a commercial device. In particular, the R-to-R wave internal (RRI) data collected with the single-arm ECG setup shows a strong correlation with that collected using the commercial device, hence validating its usefulness.

The introduction provides a very good overview of the work and the conclusions are supported by the results. The methodology and results however have several missing important parameters that limit the paper’s suitability to be published in its current form, and the authors are encouraged to address the issues, outlined below:

1. It appears that the overarching goal of the paper is to develop an ECG measurement setup that would allow for the continuous monitoring of a subject’s physiological parameters as they go about their normal daily active activities, i.e. walking, moving objects, etc, as opposed to just resting. However, this aspect of “normal daily activities” is not well emphasized in the paper, as it ought to. Limb-based ECG setups (as in the gold standard used in this work) are also capable of continuous monitoring, but inconvenient to use when doing daily activities, thus this difference should be clearly accentuated.

2. The paper’s results will be more practically useful and compelling if in addition, the authors consider including data with the subject walking and moving about, as would be the case in real life environment.

3. The need for the development of an ECG device for single-arm measurements is not clear. Why can’t the same devices used for Lead I ECG recording or similar kind be also used for single-arm setups? Did the authors try this and find the results not as good as their home-made device? Please include some clarification in the paper addressing this.

4. Holter monitors are popular as wearable ECG recorders. Please comment on their electrode placements and measurement setup in relation to yours in terms of comfort, easy of use, and performance. What are the advantages and disadvantages if any, are there any limitations to these monitors that make them not as good for monitoring heart activity when performing one’s daily activities?

Also, with your device, how would the electrode wires or cables be handled for a person wearing an apparel that covers the arms, say, a long-sleeved shirt ? Holter monitors have their electrodes placed on the chest and thus easily allow such dresses to be worn over them. Please provide more details on the practical implementation of your setup.

5. Fig.3 illustrates two different electrode placement positions for the single arm. Which one of the two was used in this study?

6. In addition to the train of ECG waveforms presented, please also include a single waveform (showing only one P wave, T wave, and QRS complex) so the features are clearly seen.

7. Mention the activity from which the waveform in fig.5 was taken from.

8. There is some concern that the selection of all-male participants may make the paper’s results biased and not representative of the general populace. Please comment on this.

9. In regard to the first sentence of the discussion section, has it been verified that the positions of the electrodes were solely responsible for the lower quality of results from the single arm? Could the single-arm ECG device itself (i.e. electronics) have any limitations that may have contributed to the lower quality? Also, was this device’s performance first validated with a clinical-grade ECG device to determine its standard? This is important as it bears a lot on the results obtained.

10. Table2 has a wide discrepancy between the SNR values of different subjects. For example, the resting activity has SNR as low as 5.6 dB and as high as 79.3 dB. What is the cause of this and is the data collection and values repeatable and consistent for the same person and same activity?

11. The number of total RRIs reported is 9568. How does this number break down into 7 activities with 30 subjects?

12. Fig.6 is blank. Please revise.

13. Table4 description ( in lines 219 – 223 of manuscript) is inaccurate. It should be that the best R was from the resting state with 0.24 value and worst was the Valsalva with 0.172 value. The second paragraph of the discussion section also refers to the wrong figures. Please re-read and verify the entire manuscript for accuracy.

English quality is mostly good, but paper has some grammatical errors that need correcting. 

Reviewer 2 Report

no specific comments

no specific comments

Author Response

To Reviewer #2:

Thank the first reviewer for his/her valuable comments that make better this manuscript. The texts in this revised manuscript have been corrected/ modified by red words. It is our sincere hope that this revision will enhance readability and strengthen of the manuscript to satisfy the requirements of this prestigious journal.

ANS: Many thanks for reviewer’s comment.

Reviewer 3 Report

Lines 36-51: Doesn't converge to HRV topic. Focus more on the second part of Introduction where benefits of HRV monitoring are presented in the context of wellbeing.

Lines 93-95: HRV is measured also by PPG technique which is already implemented into wearable devices like smartwatches. Advantages of HRV via ECG vs PPG shall be presented here.

Lines 111-118: From Figure 1, seems that ECG lead 1 is used as the reference signal. It shall be described this statement.

Figure 1: please increase the font size of D(LA) and F(REF) labels.

Figure 2: please increase the font size for resistance labels.

Figure 6: is blanck. No graph is plotted.

Figure 7: "Mean" labels font shall be increased and moved apart the lines. Hard to read in the current form.

Lines 297-306: According to the arm ECG design, another possible limitation is represented by the small distance between electrods. This should be compared to the actual wavelength of ECG wave which develops on the surface of the skin. A longer wavelenght which is greater than the physical distance of the measurements points will effect the recorder waveshape.

Round 2

Reviewer 1 Report

Authors have provided a comprehensive revision to the original manuscript, and satisfactorily addressed all comments, hence recommends to the editor that the revised manuscript be published.

The quality of English language still has some room for improvement.